# Development of Superabsorbent Polymer (SAP) Seed Coating Technology to Enhance Germination and Stand Establishment in Red Clover Cover Crop

Masoume Amirkhani *, Hilary Mayton , Michael Loos and Alan Taylor *

Cornell AgriTech, School of Integrative Plant Science, Horticulture Section, Cornell University, Geneva, NY 14456, USA
* Correspondence: ma862@cornell.edu (M.A.); agt1@cornell.edu (A.T.)

**Abstract:** Drought conditions after sowing threaten the seedling establishment of all seeds, including cover crops. Cover crops are commonly broadcast and, thus, are often susceptible to drought stress after sowing. Our hypothesis was that seed coating with superabsorbent polymers (SAPs) would enhance germination in the lab and stand establishment in the field by increasing water availability to single seeds. Red clover (*Trifolium pratense* L.) seeds were coated with the following selected SAP formulations at 2% of their seed weight: cross-linked potassium polyacrylate (PAL), cross-linked polyacrylamide-based polymer (PAM), PAM with graphite (PAM+G), and Starch-g-2-Propenoic acid (potassium salt) (STR). The water absorbency of each SAP formulation was >200 g water/g of polymer; STR had the greatest absorbency, at 352 g water/g of polymer. A seed coating method was developed, resulting in the uniform application of SAP from seed to seed. All SAP coating treatments increased germination compared to the 0% SAP coating in controlled environment studies in the lab. Three field trials were conducted for each seed coating treatment, providing a range of climatic soil conditions. Within each field trial, the STR with the greatest water absorbency had a higher stand for treatments sown by broadcasting followed by raking to incorporate seeds. The first two trials were conducted under more stressful conditions. PAM+G performed best in the first two trials by broadcasting seeds with no raking. Collectively, the selected SAP seed coating improved field stands compared to the non-treated controls.

**Keywords:** seed technology; hydrophilic polymers; water deficit; germination



## 1. Introduction

Drought is one of the most severe environmental stresses for plant productivity. It can limit agricultural production in arid and semi-arid regions, and this effect has been intensified by climate change. Cover crops are commonly broadcast over the soil surface either several weeks after the primary crop is sown or after harvest, and they often encounter transitory drought stress after seeding, which reduces stand establishment.

Superabsorbent polymers (SAP) are highly absorbent materials used in hygiene products such as disposable diapers and menstrual pads, but they also have a long history of use in agriculture and the horticulture industry [1–4]. Application of SAP to soil has beneficial results for soil rehabilitation and soil water retention in semi-arid and arid areas [5,6]. When mixed into soil at concentrations of 0.2% and 0.4% of the soil weight, SAP increased the water-holding capacity of the soil, leaving more water available for trees, and increased the tree survival of nine tree species after the initiation of drought conditions [7]. However, the soil applications of SAP are expensive. To increase the economic efficiency of SAP applications, several seed companies developed methods to apply SAP directly as a seed coating, increasing the water availability for seeds and seedlings [8,9]. SAP seed coatings were shown to increase germination and stand establishment at substantially lower application rates than soil-applied SAP [10–17].

Nitrogen-fixing legume cover crops, such as red clover, present an opportunity for farms to increase forage production while also improving soil quality and environmental sustainability [18]. However, a recent survey indicated that under 10% of New York's dairy farms utilize red clover (*Trifolium pratense* L.) cover crops [19]. This low rate of adoption may be due to the challenges associated with the successful establishment of cover crops. Unlike more cold-tolerant cover crop species such as cereal rye, red clover must be seeded in the late summer or early fall to successfully establish it before winter and produce adequate spring biomass before termination [18]. If SAP seed coatings were able to alleviate the inconsistent establishment of broadcast or inter-seeded cover crops, red clover cover crops would produce more benefits for New York farms and the environment at a cost similar to that of other late-seeded cover crop species.

SAP seed coatings may also provide some secondary benefits to facilitate broadcast sowing and seed placement accuracy for cover crops. According to the review by Wilson et al. [20], which focused on aerial seeding of cover crops, "At this point, aerial seeding is more art than science". This is largely due to the complex wind speed and seed aerodynamic factors that can influence the accuracy of broadcast seeding. Additionally, they found that heavier seeds may be less vulnerable to wind-related drift, resulting in higher seed placement accuracy. Seed placement accuracy is essential for a consistent cover crop stand and may allow growers to utilize lower seeding rates. Although the actual seed weight increase provided by SAP coating will be dependent on the type of seed, as well as the density and build-up of the coating applied, it is likely that coated seeds will benefit broadcast seeding accuracy [20,21]. The type of seed treatment, as well as environmental variables such as high humidity, can impact the seed flow characteristics. The polymer application rate, surface texture, and lubricity of a seed coating all influence seed flowability. Therefore, accurate measurement of seed flow is critical in seed coating development. The more seeds (in number or weight) to pass through the seed flow meter or planter for a given time, the better the flowability [22,23].

Most studies analyzing the impact of SAP seed coatings on germination and stand establishment focused on native plant species seeded into natural environments to restore native sites [24], with no known studies analyzing the impact of SAP seed coatings on cover crop germination or stand establishment. Moreover, published field studies on SAP seed coatings include little supporting lab data. The objectives of this study were to (1) quantify the water absorbency of the selected commercial SAP formulations; (2) develop a seed coating method for the uniform application of SAP as a component of the seed coating; (3) examine seed flow properties of SAP-coated seeds; (4) compare the efficacy of SAP-coated seeds by controlled moisture studies in the lab; and (5) conduct field studies of SAP-coated seeds sown by broadcasting or incorporated via raking the at time of sowing. Red clover was used as the model cover crop in all studies to test the hypothesis that seed coating with superabsorbent polymers (SAP) would enhance germination in the lab and stand establishment in the field by increasing water availability.

## 2. Materials and Methods

### 2.1. SAP Materials and Determination of Water Absorbency of SAP Materials

Four superabsorbent polymer (SAP) formulations were tested in this study: cross-linked potassium polyacrylate (PAL), cross-linked polyacrylamide-based polymer (PAM), PAM with graphite (PAM+G), and Starch-g-2-Propenoic acid (potassium salt) (STR) (Table 1). The abbreviations PAL, PAM, PAM+G, and STR will be used throughout the paper.

**Table 1.** Hydrophilic commercial polymers used in this study.

| Treatment | | Component | Company and Location |
|---|---|---|---|
| 1 | CONTROL | | |
| 2 | 0% * SAP | | |
| 3 | 2% SAP (Stockosorb-90AC) (PAL) | Cross-linked potassium polyacrylate | Evonik Corporation 2401 Doyle Street Greensboro, NC |
| 4 | 2% Soil Moist (PAM) | Cross-linked polyacrylamide-based polymer | JRM Chemical Inc. 4881 Neo Parkway, Cleveland, OH |
| 5 | 2% Soil Moist + Graphite (PAM+G) | Cross-linked polyacrylamide-based polymer | JRM Chemical Inc. 4881 Neo Parkway, Cleveland, OH |
| 6 | 2% K-BOOST STARCH (STR) | Starch-g-2-Propenoic acid, potassium salt, polymer | D2 Polymer Technologies Inc. 1539 N Wooster Road Scottsburg, IN |

* SAP = Superabsorbent polymer.

The volumetric liquid absorption method was used to determine the water absorbency of different SAP formulations. The methods were adapted from Evonik Corp [25]. First, 3 replicates of 0.5 g of each SAP were slowly poured into 200 mL of RO/DI water while stirring. After 15 min of stirring and a resting time of 1 h 45 min, the liquid was filtered with a 100-mesh screen. After a draining time of 20 min, the remaining amount of liquid was determined.

### 2.2. Seed Coating Method

A seed coating technology was developed at Cornell AgriTech to apply selected hydrophilic polymers (Table 1) using red clover seeds (*Trifolium pratense* L., "VNS-variety not stated", provided by King's AgriSeeds, Inc., Lancaster, PA, USA). A commercial seed coating filler and binder combination (Kbind filler and KrustMaster binder) was provided by Kannar Earth Science Ltd. (Lawrenceville, GA, USA). A laboratory-scale rotary pan coater with a diameter of 15 cm, R-6 (Universal Coating Systems, Independence, OR, USA), was used to coat the red clover seeds. For each coating batch, the powder and liquid were applied to the surface of the seeds in incremental amounts as they rotated in the R6 to achieve uniform results. A 1:1 ratio of seed to filler was used. The coating technology was an encrustment [17], but will be termed coated seed or seed coating throughout the study. To improve our ability to observe seed coating uniformity, a red dye (Pro-Ized red colorant, Bayer, Research Triangle Park, NC, USA) was added at a rate of 10% to the liquid binder solution. After coating, the seeds were dried at room temperature for 48 h until completely dry. As hydrophilic polymers rapidly swell and can become sticky during coating, there were three unique steps employed for successful seed coating. The dry polymer formulations were sieved through a 200-mesh (<74 microns) or smaller sieve to remove larger particles, as a finer particle size improves the uniformity of the application due to lower seed-to-seed variation. The finely sieved polymer powders were thoroughly mixed with the filler. Finally, the seed treatment binder was a viscous liquid which was diluted with sufficient water to allow for good atomization by the R6. It also prevented the moving water from rapidly hydrating the polymer during seed coating.

### 2.3. Determination of Flowability of SAP Coated Seeds

To test the flowability of SAP-coated seeds, a laboratory-scale seed flow meter (Centor Oceania's Lab FlowTek, Carrum Downs, Victoria, Australia / Aginnovation LLC, Walnut Grove, CA, USA) was used (Figure 1a,b). Seed flow meters measure how quickly seeds will pass through planting equipment, and predict problems with lodging in hoppers or reductions in seed flow related to the coating materials compared to the non-treated seeds. It also helps with gauging plantability in the field. For this study, we modified the seed

flow meter for use with small seeds. A reducer with 6 mm diameter opening was machined from high-density polyethylene (HDPE) and inserted into the hopper (Figure 1c).

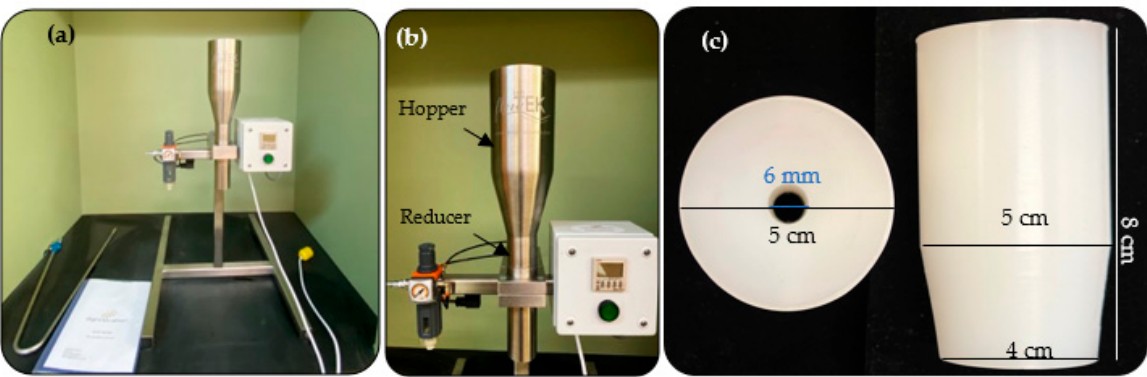

**Figure 1.** (**a**) A seed flow meter (Centor Oceania's Lab FlowTek, Victoria, Australia) was used to determine the seed flowability. (**b**) Close up of hopper and where the reducer inserts. (**c**) The reducer was custom-made in Cornell AgriTech from High-density polyethylene (HDPE) and modified the seed flow meter; its use is feasible for small-seeded crops in small batches of 25 g.

To determine the effect of the increase in SAP in the coating material on seed flowability, red clover seeds were coated with 0, 1, 2, 4, 8, 16, and 20% PAL. These were then passed through the seed flow meter and measured (g seeds/0.5 s) for all treatments. The flowability data were also recorded for the 2% PAL-, PAM-, PAM+G-, and STR-coated seeds and compared with the 0% SAP and non-treated control seeds.

### 2.4. Laboratory Investigations of Coated Seeds

To assess the effect of the SAP coating on seed germination under water deficit conditions, two different experiments were designed. The first was limited water to single seeds. In this experiment, to test each coating treatment as well as whether the SAP formulation restricted water availability to seeds, a limited amount of water (6, 8, and 10 µL) was pipetted onto a single coated seed for each treatment (4 reps of 25 seeds) and kept at 10/20 °C. Fine mesh screens (10 × 10 cm$^2$) were used to support seeds and restrict the water's movement to adjacent seeds. Germination was recorded 48 h after planting. For simulated drought stress, 4 replicates of 50 seeds of each treatment, including non-coated control seeds, were broadcast on MS 40/100 LVM montmorillonite clay (Oil-Dri Corporation of America, Chicago, IL, USA) with 95% moisture content and kept at 20 °C in the germinator. The montmorillonite clay was placed in rectangular plastic containers with lids. After 40 h, the moisture content (MC) of the media decreased to 45% by evaporation, on average, for 4 replicates. At that time, the containers were closed to maintain an adequate moisture content for germination. Germination was recorded and cumulative germination percentages were calculated over 7 days. Germination velocity, estimated by using Timson's index [26], was calculated for each treatment as follows.

$$\Sigma k = \sum_{i=1}^{k=7} G i$$

where $Gi$ is the cumulative germination percentage in time interval $i$, and $k$ is the total number of time intervals.

### 2.5. Field Studies of Coated Seeds

Three field trials were conducted in 2020 with six seed treatments: non-coated control, coated control, and four coating treatments: PAL, PAM, PAM+G, and STR. A randomized complete block design was used for all trials. All six treatments were broadcast alone (B) or broadcast and then incorporated into the soil by raking (BR), resulting in twelve

total treatments. Broadcast (B) seeding is a method of planting that involves scattering seeds over the plots. For the broadcast and raking (BR) method, broadcasted seeds were lightly incorporated with a rake to improve seed-to-soil contact. All 12 treatments were sown on 26 June, 24 July, and 17 September, respectively, which allowed us to conduct the experiment in variable environments (Figure 2a). Field location, planting and final stand count date, soil type, and weather data are shown in Tables S1 and S2. For all treatments, 150 seeds were sown per plot ($30 \times 60$ cm$^2$) (Figure 2b) with 6 replications for all 3 trials. For both planting methods (B and BR), to distribute the seeds uniformly across the plots, a Vibro Hand Seeder (GRO-MOR Inc., Adams, MA, USA; http://gro-morent.com/gro-mor.htm accessed on 15 May 2020) was used at the time of sowing. For an even distribution of seeds, the speed was adjusted to medium for all treatments. Stand counts were recorded starting at day 10 and continued every 10 days for 1 month after planting for each trial. Stand counts were based on above-ground emergence of the cotyledons of red clover seedlings.

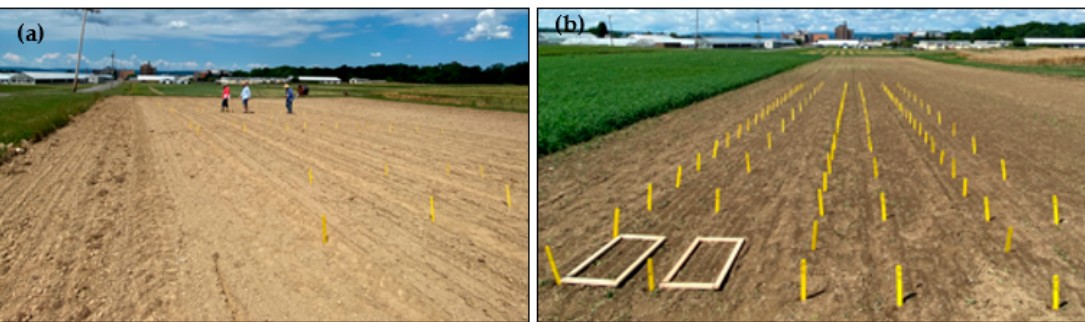

**Figure 2.** (**a**) Trial 1, planted under warm and dry conditions. (**b**) Trial 2, illustrating the planting wood template, with an area of $30 \times 60$ cm$^2$.

### 2.6. Statistical Analysis

Analysis of variance (ANOVA) ($\alpha = 0.05$) and Tukey's HSD (honest significant difference) tests were performed and analyzed by JMP$^{\circledR}$ Pro 14 (SAS Institute Inc., Cary, NC, USA) for polymers' water absorbency, seed cumulative germination, and field stand count data [27]. All graphs were drawn with Microsoft Excel 2021 (Microsoft$^{\circledcirc}$, Redmond, WA, USA).

## 3. Results and Discussion

### 3.1. Determination of Water Absorbency of SAP Materials

Each SAP material requires free water from the environment to hydrate and swell. The water absorbency (g water per g polymer) levels of different SAPs were determined with three replicates using the volumetric liquid absorption method (Figure 3a). Our findings demonstrated that all SAP formulations had water absorbency levels greater than 200 g water/g SAP (Figure 3b). STR held a significantly higher amount of water (352 g water/g STR), followed by PAM, PAM+G (260 g water/g PAM/PAM+G), and PAL (215 g water/g PAL) (Figure 3a). The molecular size and the degree of crosslinking had a great impact on the water absorbency of SAP [28]. In another study, the molecular weight of cornstarch was reported to be $10^6$ gmol$^{-1}$, while the different polyacrylamide polymers (PAM) were $3 \times 10^5$ to $4 \times 10^5$ [29]. The amount of water absorbed by STR particles was similar to Zeba$^{\circledR}$, a commercial cornstarch-based absorbent powder that can absorb more than 400 times its original weight in water [30]. Water absorbency values from PAL, using double-distilled water and longer soaking periods, were recorded for two trials as 272 and 292 g water/g [31]. Therefore, both the water quality and the test method may have altered the measured water absorbency.

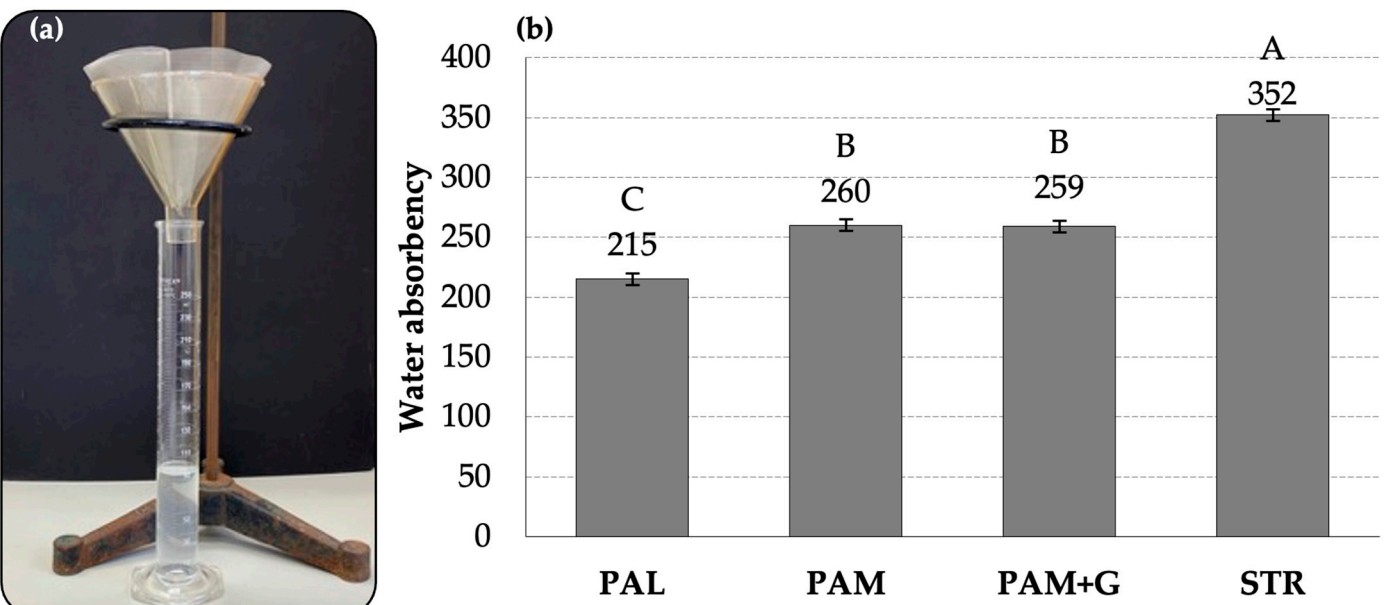

**Figure 3.** (**a**) The volumetric liquid absorption method used to determine the (**b**) water absorbency (g water per g polymer) of different SAPs. Different letters indicate significant differences among water absorbency of different polymers ($p < 0.05$).

### 3.2. Seed Coating Method

Seeds were coated (2% of the seed weight) with four commercially available SAP formulations (Figure 4a) using a laboratory-scale rotary pan coater (Figure 4b). The uniformity of the SAP application was observed by the swelling of individual SAP-treated seeds (compare Figure 4c with Figure 4d). To further display the rapid water uptake of the SAP-coated seed, a time-lapse video (Video S1) shows the 20% PAL coating swelling over a 60 s period. Throughout this study, coated seeds placed on moistened blue blotters (Anchor Paper Co., St. Paul, MN, USA) were used to assess the application accuracy of SAP on single seeds (Figures 4c,d and 5a,c; and Video S1). Collectively, the uniformity of the SAP application was achieved by observing the swelling of single seeds on moistened blotters.

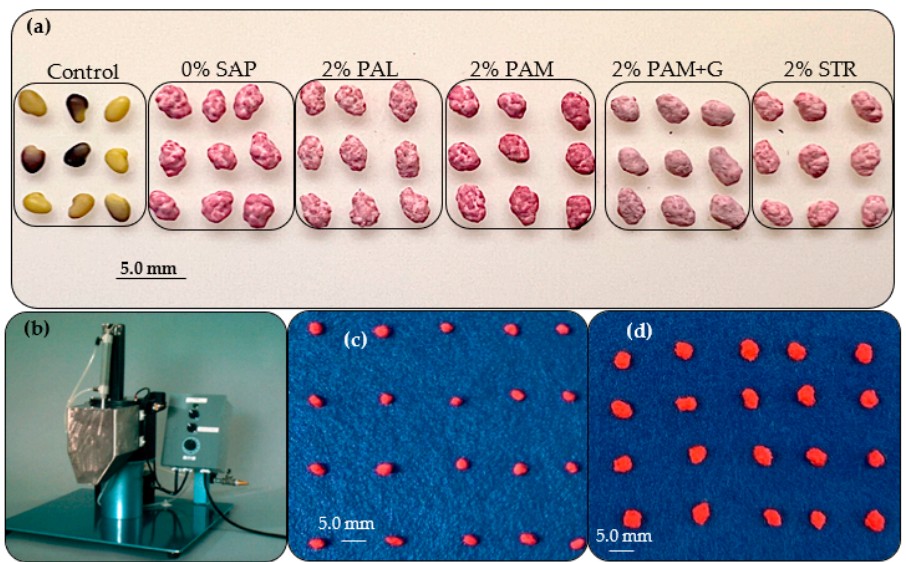

**Figure 4.** (**a**) Non-treated control and coated red clover (*Trifolium pratense* L.) seeds used in this study (Table 1). (**b**) Rotary pan coater-R6. Comparison of seeds coated with (**c**) no SAP (0% SAP) and (**d**) 2% PAL coating (PAL coating held moisture around seeds).

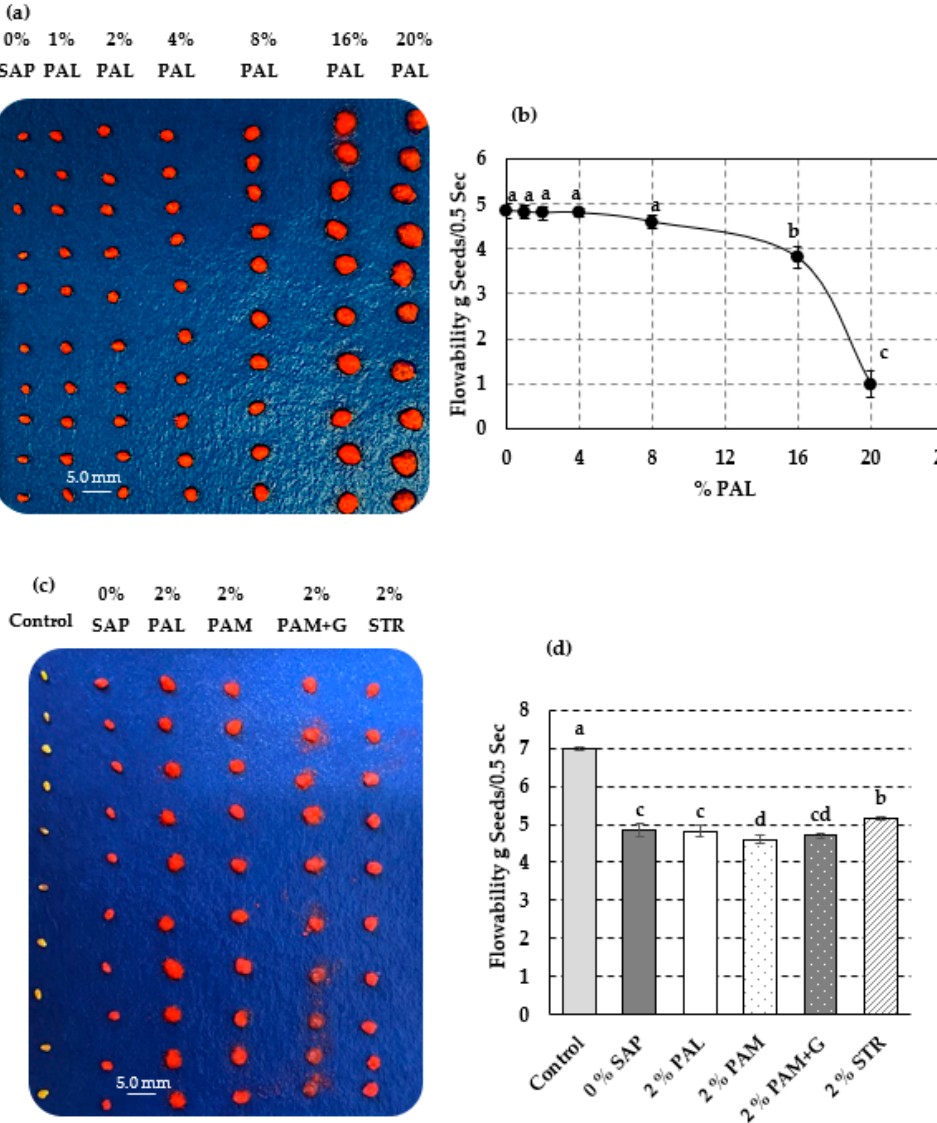

**Figure 5.** Effect of increasing PAL% in coating materials on coated seeds' (**a**) water absorbency and (**b**) flowability performance. Comparison of 2% SAP-coated seeds' (**c**) water absorbency and (**d**) flowability performance with the coated seeds without SAP and the non-treated control red clover seeds. Different letters above error bars indicate significant differences among treatments ($p < 0.05$).

*3.3. Seed Flowability of the SAP-Coated Seeds*

Although the increase in PAL % in seed coatings increased swelling (Figure 5a), it had a significant negative impact on seed flowability (Figure 5b). PAL > 8% decreased seed flowability, and seed flowability decreased to 3.8 and 1 g seeds/0.5 sec at 16 and 20% PAL, respectively. Increasing PAL from 1–4% to 20% decreased the seed flow rate by approximately 80% (Figure 5b). It is likely that a higher percentage of the polymers absorbed moisture from the air, resulting in surface adhesion of the coated seeds and thus lowering flowability.

Figure 5c demonstrates fully hydrated, coated seeds with both 0 and 4 2% SAP formulations in comparison with the non-treated control red clover seeds. Through this observation, we concluded that all SAP-coated seeds received polymers during the coating operation to cause them to absorb water and swell immediately after contact with the free water on the blue blotter. Independent of the formulation of the coating materials, all coated seeds had lower flow rates than the non-coated seeds (shown in Figure 5d). However, STR-coated seeds had slightly higher flowability compared with the other treatments. These

results show that both the formula and amount of polymers used in seed coating affect seed flowability and, therefore, ease the process of planting in the field. Results from other research have revealed that corn seed flowability was enhanced by approximately 50% when a polymer + active film coating was used compared to an active-only coating on film-coated corn seeds [22]. Pawlicki et al. (2019) [32] examined three types of commercially coated seeds with low, moderate, and high flowability to investigate seed flow challenges related to agrochemical polymeric coatings. Their results showed that adhesion is driven by the selection of a polymer matrix which is superior in controlling the microscale distribution of chemistry and, therefore, overall flowability [32].

*3.4. Laboratory Investigations of Coated Seeds*

Red clover seeds coated with SAP formulations resulted in better germination and radical emergence, independent of the limited water available (6, 8, and 10 µL) per seed after 48 h. Under all conditions, seeds treated with SAP had higher germination percentages when compared to coated seeds with no SAP in the coating formula (Figure 6). However, when increasing water availability from 6 to 10 µL per seed, germination dramatically increased for all treatments, including 0% SAP, for which it increased from 26 to 66%. Collectively, 2% application of these polymers, regardless of chemistry, effectively increased the germination to >75% for 8 µL and >90% for 10 µL water available per seed (Figure 6). Moreover, none of the SAP chemistries restricted water availability for germination.

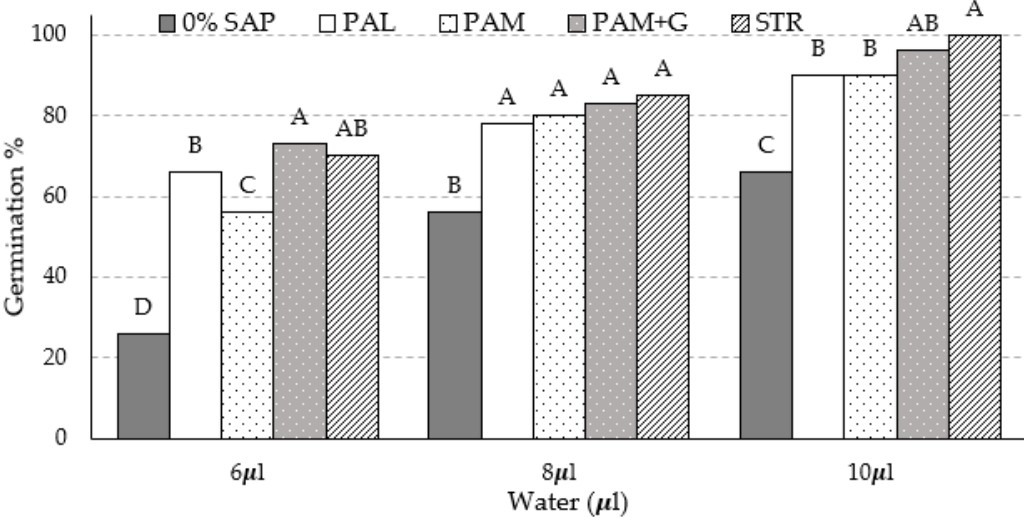

**Figure 6.** Germination % after 48 h with controlled hydration (6, 8, and 10 µL water added per seed) at 10/20 °C. Different letters indicate significant differences for each available water level ($p < 0.05$).

To simulate drought stress during germination, all six treatments were sown under controlled conditions in Montmorillonite clay media. The media was air-dried after sowing, and the containers were sealed after 40 h (MC 45%). One week after broadcasting in Montmorillonite clay media, total germination was 67% for the non-treated controls and 60% for the 0% SAP group. To compare, germination was 93–98% for the red clover seeds coated with all 2% SAP treatments (Figure 7). The Timson's indices [26] for germination velocity in the non-treated control, 0% SAP, PAL, PAM, PAM+G, and STR groups were 169, 118, 282, 288, 283, and 284, respectively (where greater values have higher and faster germination rates). Collectively, all 2% SAP formulations had similar Timson's index values, ranging from 282 to 288, compared to the non-SAP checks.

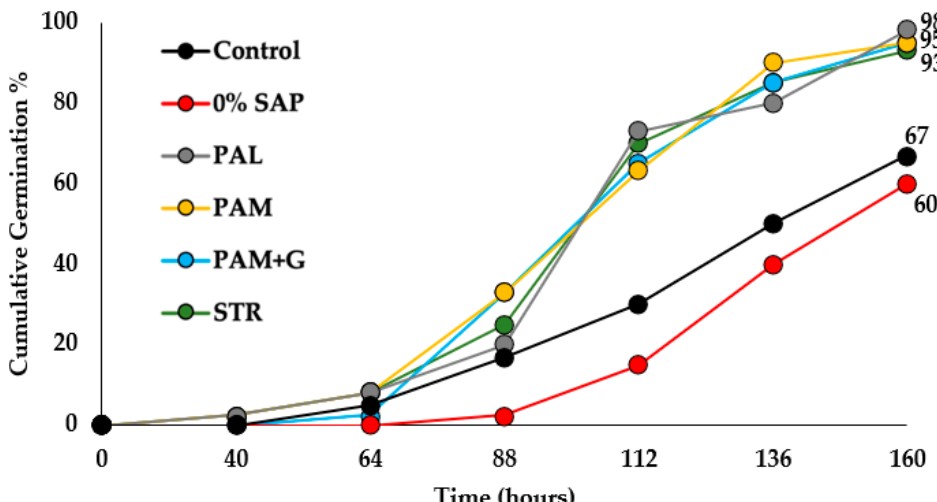

**Figure 7.** Cumulative germination % of 6 treatments of red clover seeds, broadcasted on montmorillonite clay at 20 °C.

Collective data from the two lab studies verified that all SAP seed coatings improved germination compared to the 0% SAP-coated seeds in controlled moisture environments. Moreover, none of the SAP chemistries restricted water availability, nor did they negatively impact germination.

In a laboratory study, Tao et al. [33] showed that adding carbohydrate-based superabsorbent polymers to maize seeds enhanced the germination potential, germination rate, and total biomass of corn seedlings. Laboratory studies were also conducted by Baxter and Waters [11] to determine the effect of the hydrophilic polymer Waterlock B100, an absorbent starch polymer, on the imbibition and germination of sweet corn seeds at soil/water matric potentials of −0.01, −0.40, −1.0, and −1.5 MPa. The polymer-coated seeds had a higher final percentage of imbibition and higher rates of respiration and germination, at −0.01 and −0.40 MPa, than uncoated seeds. However, as the water potential decreased below −1.0 MPa, the seed coatings had a detrimental effect on the physiological processes leading to germination.

### 3.5. Field Studies of Coated Seeds

The mean seedling emergence for each of the field trials sown on 26 June, 24 July, and 17 September is shown in Figure 8a. The first two field trials had 29 and 43 mm precipitation for Trial 1 and 2, respectively (Table S2). Field Trial 3 was planted under slightly cooler day and cold night temperature conditions, and precipitation was 63 mm (Table S2). Therefore, soil water availability had a major effect on stand establishment. For all treatments, regardless of planting date, the raking method which employed enhanced seed–soil contact, resulted in higher seedling stand counts and growth parameters compared with broadcasting only (Figure 8b and Table 2).

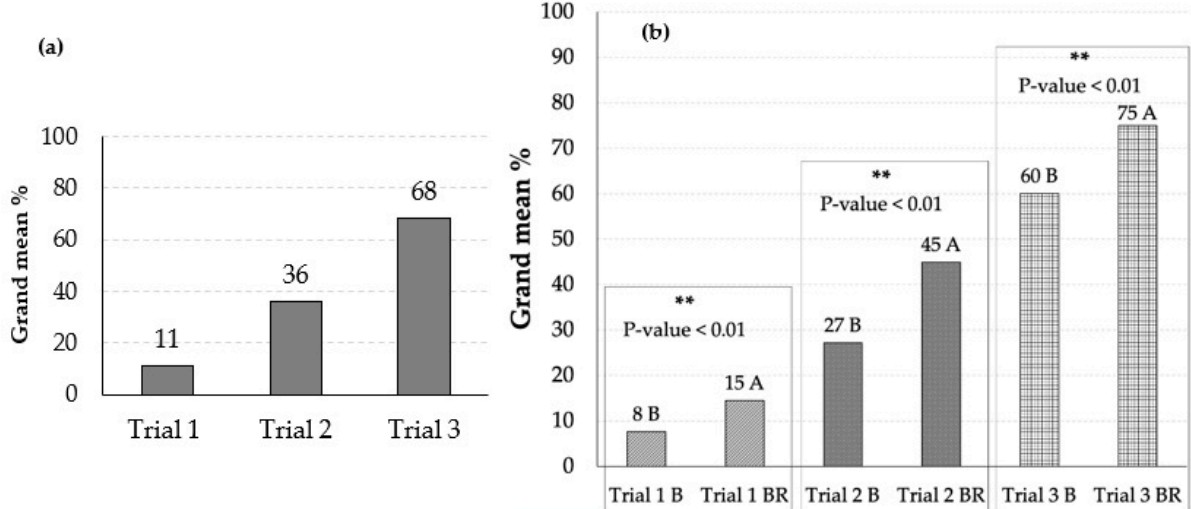

**Figure 8.** (**a**) The grand mean seedling emergence % (average of all 12 treatments for each trial). (**b**) The mean for each planting method for each trial (average for 6 coating treatments), B = broadcasting and BR = broadcasting and raking, different letters indicate significant differences (** $p < 0.01$).

**Table 2.** Percent plant stand count of six different treatments of two planting methods (B and BR) at three planting dates in the field (B = broadcasting and BR= broadcasting and raking).

| | Stand Count % | | | | | |
|---|---|---|---|---|---|---|
| | Trial 1 | | Trial 2 | | Trial 3 | |
| **Treatments** | **B** | **BR** | **B** | **BR** | **B** | **BR** |
| Control | 6 c | 13 bc | 22 c | 40 c | 56 cd | 70 bc |
| 0% SAP | 7 c | 13 bc | 26 bc | 43 bc | 51 d | 70 bc |
| PAL | 9 bc | 15 B | 30 ab | 44 bc | 65 ab | 76 ab |
| PAM | 6 c | 13 bc | 24 bc | 45 ab | 58 cd | 73 abc |
| PAM+G | 10 ab | 13 bc | 34 a | 47 ab | 62 bc | 79 a |
| STR | 8 bc | 20 a | 24 bc | 52 a | 67 ab | 79 a |

Within a column, mean values with different letters are statistically different ($p < 0.01$).

SAP coatings affected seedling establishment for both the broadcasting (B) and raking (BR) sowing methods. Red clover seeds treated with starch (STR) had consistently higher final stand counts across all three trials with the raking method, whereas seeds treated with Soil Moist [TM] with graphite (PAM+G) showed higher rates of seedling establishment with the broadcasting method compared to other treatments sown by this method during trial 1 and 2. The average of three plantings revealed a 45% increase in stand counts for PAM+G compared to the non-coated control for the broadcast (B) method, while STR had 31% greater stand counts than the non-coated control for the incorporated method (BR) (Table 2). Further, STR had the greatest water absorbency (Figure 5d), which may have contributed to enhanced field establishment, as it was conducted in a more stressful soil environment due to low precipitation (Figure 8a). Interestingly, PAM+G had a greater stand than PAM only for the broadcast sowing method in the first and second field trials (Table 2). Both PAM formulations had similar water absorbency rates (Figure 5d), so the improved stand establishment was attributed to the graphite. This presents an opportunity for the addition of other compounds to be applied in seed coatings along with SAP to further enhance stand establishment.

The effects of the hydrophilic polymer Waterlock B100 were studied by evaluating the field performance of sweet corn and cowpea [34]. Polymer-coated seeds (1.1, 2.3, 4.6, and 9.1 g/kg seed) were planted to assess stand establishment and plant growth. Sweet corn showed significant positive outcomes due to the coatings with 2.3 and 4.6 g/kg

polymer/seed. However, all treatments had an unfavorable effect on cowpea germination and seedling development.

The results show that SAP seed coatings provide some measured benefits to both methods of planting, increasing stand establishment under a wide range of climatic conditions when compared to non-treated and treated seeds with no SAP. Seed technology, including seed coatings with hydrophilic materials, is one approach for increasing crops' tolerance to soil moisture content fluctuation after broadcast seeding.

## 4. Conclusions

The development of an effective delivery system for superabsorbent polymers via seed coatings for cover crops was achieved. Novel seed coating technologies using SAP chemistries could potentially represent one strategy to minimize transient drought stress and enhance seed germination. Collectively, the environment for each field trial had a major influence on field stand establishment, while the selected hydrophilic seed coatings significantly improved stand counts within either broadcast or soil-incorporated planting methods. Future considerations may include the use of SAP seed coatings on other crop species, or crop seeds with epigeal germination, to study the benefits of retaining SAP underground after germination. Natural hydrophilic polymers could be developed in a seed coating formulation for organic production. Therefore, continued research is warranted to assess the potential applications of SAP coatings in agriculture, especially in rain-fed (non-irrigated) agricultural farming.

**Supplementary Materials:** The following supporting information can be downloaded at: https://www.mdpi.com/article/10.3390/agronomy13020438/s1, Video S1: Time-lapse video showing 20% SAP coating swelling over a 60 s period; Table S1: Field location, soil type, planting date, and final stand count date; Table S2: Weather conditions of the three field studies after planting. Data show the first 10 days after planting for Trial 1 and 2, and 15 days for Trial 3, due to cold night conditions.

**Author Contributions:** Conceptualization, M.A. and A.T.; methodology, M.A. and A.T.; software, M.A.; validation, M.A., H.M. and A.T.; formal analysis, M.A.; investigation, M.A., H.M. and A.T.; resources, A.T.; data curation, M.A., H.M. and M.L.; writing—original draft preparation, M.A.; writing—review and editing, M.A., H.M., M.L. and A.T.; visualization, M.A. and H.M.; supervision, A.T.; project administration, A.T.; funding acquisition, A.T. All authors have read and agreed to the published version of the manuscript.

**Funding:** This material is based upon work that is supported by the National Institute of Food & Agriculture, United States Department of Agriculture, Federal Capacity Hatch Funds, under Accession #1017599.

**Acknowledgments:** We would like to thank Ben Lehman for his assistance in this project. We are thankful for the technical advice from several companies donating materials for this project: Evonik Corporation, JRM Chemical, and Kannar Earth Science.

**Conflicts of Interest:** The authors declare no conflict of interest.

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
