# Peer review of "Development of Superabsorbent Polymer (SAP) Seed Coating Technology to Enhance Germination and Stand Establishment in Red Clover Cover Crop"

_agronomy, doi:10.3390/agronomy13020438_

Round 1

Reviewer 1 Report

The authors needs to explain very well about the hypothesis of the study. In introduction part it is ambiguous , therefore need clarity. The methodology also needs revisions, the design of experimentation was not adequately explained in methodology , also the justification of various observations also not properly defined. The treatment combinations are so simple. Even barring the treatments the statistical considerations must be adequately met in the study to justify preciseness of the  finding .

The result part also need more data in tabular form. There is need to present the relevant data as per the objective of the study and the hypothesis. The biggest deficit in the manuscript was felt in results part is complete absence of soil moisture data. The use of super absorbent polymers and their response largely depend on availability of soil moisture, at various stage what was the soil moisture status  need to be very clearly presented in the result part.

Then the discussion par also need robust scientific reasoning. 

Author Response

Response to reviewer #1: Authors of this manuscript acknowledge general comments and suggestions of our manuscript. Our response (in red) follows:

Reviewer 1:

Comments and Suggestions for Authors

The authors needs to explain very well about the hypothesis of the study. In introduction part it is ambiguous, therefore need clarity. The methodology also needs revisions, the design of experimentation was not adequately explained in methodology, also the justification of various observations also not properly defined. The treatment combinations are so simple. Even barring the treatments the statistical considerations must be adequately met in the study to justify preciseness of the finding.

The result part also need more data in tabular form. There is need to present the relevant data as per the objective of the study and the hypothesis. The biggest deficit in the manuscript was felt in results part is complete absence of soil moisture data. The use of super absorbent polymers and their response largely depend on availability of soil moisture, at various stage what was the soil moisture status need to be very clearly presented in the result part.

Then the discussion par also need robust scientific reasoning. 

The stated objectives of the study are now listed at the end of the Introduction. The authors wish to make the point that the first four objectives were conducted in laboratory and provide new information in the literature on the topic of SAP seed coating technology:

  • quantify water absorbency of selected commercial SAP formulations The method was adopted from a SAP industry collaborator and our paper provides new data published on water absorbency of four selected SAP formulations.
  • develop a seed coating method for the uniform application of SAP as a component of the seed coating. SAP application is performed on commercial seed lots by the seed industry; however, methods are proprietary, so not available for a researcher to reproduce the coating technology. Methods are described in this paper with the application of up to 20% SAP in the seed coating conducted at Cornell AgriTech. In addition, the time-lapse video of the swelling of a SAP coated seed in the supplementary section is unique.
  • examine seed flow properties of SAP coated seeds Crucial to commercial sowing of SAP coated seed is the ability to flow unimpeded through a planter. Our use of a seed flow meter documented that the 2% SAP coatings had little effect on seed flow compared to 0% SAP coated seeds.
  • compare efficacy of SAP coated seeds under controlled moisture studies in the lab An experiment was conducted providing only a limited amount of water available to each coated seed that demonstrated that the superabsorbent polymers held water in the coating, but freely provided the water to the germinating seed. In addition, the four tested SAP formulations were not phytotoxic to germination. The second lab study provided a simulated drought by allowing the growing media to dry after planting. In this scenario, free water was available for the SAP to rapidly swell, and hold the water around the coated seed resulting in germination in a drying environment. Previous published papers on SAP coated seed do not include studies under controlled environments.

Specific points are listed in response to reviewer #1

Title revised LINE 2-4

Abstract rewritten LINE 9-24

Clear list of objectives added at the end of the Introduction LINE 75-85

Results & Discussion section expanded to connect objectives and hypotheses with results.  LINE 386-389, LINE 429-432& LINE 468-475

Laboratory investigations were conducted under controlled environmental conditions.  Positive efficacy results of SAP coatings from these lab investigations provided justification for field studies. In these field studies, air temperature and precipitation were presented in the Supplementary section. Three field were conducted to provide trials at different times of the year providing a range of climatic conditions. This wide range of climatic conditions was achieved as the mean field emergence was 11, 36 and 68%. Soil moisture was not measured as the objective was to test the seed coating efficacy within each planting date/trial and planting method. The overall consistent result of the field work was that the major influence on field stand establishment was the environment, while within each field study selected SAP seed coatings significantly improved stand counts compared to the noncoated control or 0% SAP coating.

Field data of Fig 9 of original draft now presented as Table 2: LINE 482-484

Table 2. Percent plant stand count of 6 different treatments of two planting methods (B and BR) at three planting dates in the field (B = Broadcasting and BR= Broadcasting + Raking).

Stand count %

Trial 1

Trial 2

Trial 3

Treatments

B

BR

B

BR

B

BR

Control

6 c

13 bc

22 c

40 c

56 cd

70 bc

0% SAP

7 c

13 bc

26 bc

43 bc

51 d

70 bc

PAL

9 bc

15 B

30 ab

44 bc

65 ab

76 ab

PAM

6 c

13 bc

24 bc

45 ab

58 cd

73 abc

PAM+G

10 ab

13 bc

34 a

47 ab

62 bc

79 a

STR

8 bc

20 a

24 bc

52 a

67 ab

79 a

Within a column, mean values with different letters are statistically different (P < 0.01).

The authors wish to note that the same draft of the manuscript had diverging responses and copied from Reviewer #3.

Reviewer #3 Comments and Suggestions for Authors: The methodology is easy to follow and the results are clearly presented.

Reviewer 2 Report

Dear Authors,

The abstract of the present manuscript describes the same results and conclusions than a conference abstract which is publicly available (https://ashs.confex.com/ashs/2021/meetingapp.cgi/Paper/34964). Furthermore, there are sentences and at least a paragraph that are identical to the conference abstract (e.g. Ln29-33).

I am confident that your experiments have much more information and conclusions than those already presented and published in the conference. I believe it is absolutely needed to reflect ‘new’ findings in the abstract of a manuscript to be published in a journal. Unfortunately, as your abstract do no reflect such new findings, I would not further review this manuscript in the current state and I do encourage you to do a resubmission. If you decide to do so, please, make sure to avoid using already published paragraphs, even if you are the author. Also, reflect new findings, for example, you could mention the used of SAP with increasing water absorbencies (Fig 3, PAL > PAM & PAM+G >> STR) and that lab tests resulted in higher germinantion on SAP coated seeds than noncoated and coated 0%, regardless water absorbency of SAP or the water availability (Fig. 6). Also, that the planting method had significant effects on the field results.

A final general comment is to carefully check that the references given are appropriate to backup what it is stated in the manuscript.

Below, some specific comments:

ABSTRACT

Ln10. Be consistent and use either ‘superabsorbent’ or ‘super absorbent’.

Ln14-15. ‘STR had greater water absorbency than other SAP materials’ .Is this from methods or a result? I suggest deleting from the abstract unless it is a meaningful result and conclusions are developed. If it is a relevant result, you need to present it as such.

Ln18. Please, replace ‘and 4 SAP treatments’ with '...PAL, PAM, PAM+G and STR)'. Make sure to add the closing bracket.

Ln18. Delete ‘different’ and include the methods in brackets ‘(broadcast and broadcast with ranking).

Ln22. Please, replace ‘for the broadcast’ with ‘under broadcast planting’.

Ln22-23. Please, correct to ‘while STR had 31% greater plant stand than the noncoated control under broadcast with ranking planting’.

Ln23-24. Please, replace ‘selected hydrophilic seed coating’ with ‘selected SAP seed coating’.

Please, indicate if the differences in plant stands were statistically significant by adding significance level (e.g. P < 0.05).

Please, in the keywords, do not include words already used in the title such as seed coating, cover crop, superabsorvant polymers.

INTRODUCTION

Ln29-33. This paragraph is identical to a presentation abstract publicly available and needs to be re-writing (https://ashs.confex.com/ashs/2021/meetingapp.cgi/Paper/34964).

Ln37-38. Are references 5 and 6 related to semi-arid and arid environments? If not, please, replace with appropriate references.

Ln40. Reference 7 is not appropriate. Please, replace.

METHODS

You need to describe the differences between broadcasting and broadcasting with ranking. These methods yielded significantly difference seedling emergencies (Fig. 8b).

RESULTS

Fig. 9. Apparently, the significant comparisons were conducted all across the 12 treatments. However, this seems to be inappropriate because the planting method (broadcast vs. broadcast with ranking) might confound the results. If planting method is not included in the analysis as an interaction factor, I would suggest to perform ANOVA per trail and within planting method.

REFERENCES

Please, be consistent and either include the doi link for all citation or don’t such as in reference 2: ‘https://doi.org/10.1007/BF00750062’. I suggest to include it in all references.

Author Response

Response to reviewer #2: We would like to thank Reviewer #2 for the careful and detailed reading of our manuscript and for the thoughtful comments and constructive suggestions, which help to improve the quality of our manuscript. Our response (in red) follows: 

Reviewer #2:

Dear Authors,

The abstract of the present manuscript describes the same results and conclusions than a conference abstract which is publicly available (https://ashs.confex.com/ashs/2021/meetingapp.cgi/Paper/34964).

(Authors agree with the comment and rewrote the abstract, please see line 9-24)

Furthermore, there are sentences and at least a paragraph that are identical to the conference abstract (e.g. Ln29-33). (Addressed in point 9: This paragraph paraphrased accordingly, please see line 28-32.

I am confident that your experiments have much more information and conclusions than those already presented and published in the conference. I believe it is absolutely needed to reflect ‘new’ findings in the abstract of a manuscript to be published in a journal. Unfortunately, as your abstract do no reflect such new findings, I would not further review this manuscript in the current state and I do encourage you to do a resubmission. If you decide to do so, please, make sure to avoid using already published paragraphs, even if you are the author. Also, reflect new findings, for example, you could mention the used of SAP with increasing water absorbencies (Fig 3, PAL > PAM & PAM+G >> STR) and that lab tests resulted in higher germination on SAP coated seeds than noncoated and coated 0%, regardless water absorbency of SAP or the water availability (Fig. 6). Also, that the planting method had significant effects on the field results.

A final general comment is to carefully check that the references given are appropriate to backup what it is stated in the manuscript.

The one paragraph abstract from a poster at a conference was only on field data, while this full manuscript includes several unique lab studies with field data. The revised abstract now encompasses the full findings (line 9-24).

The stated objectives of the study are now listed at the end of the Introduction. The authors wish to make the point that the first four objectives were conducted in laboratory and provide new information in the literature on the topic of SAP seed coating technology:

  • quantify water absorbency of selected commercial SAP formulations The method was adopted from a SAP industry collaborator and our paper provides new data published on water absorbency of four selected SAP formulations.
  • develop a seed coating method for the uniform application of SAP as a component of the seed coating. SAP application is performed on commercial seed lots by the seed industry; however, methods are proprietary, so not available for a researcher to reproduce the coating technology. Methods are described in this paper with the application of up to 20% SAP in the seed coating conducted at Cornell AgriTech. In addition, the time-lapse video of the swelling of a SAP coated seed in the supplementary section is unique.
  • examine seed flow properties of SAP coated seeds Crucial to commercial sowing of SAP coated seed is the ability to flow unimpeded through a planter. Our use of a seed flow meter documented that the 2% SAP coatings had little effect on seed flow compared to 0% SAP coated seeds.
  • compare efficacy of SAP coated seeds under controlled moisture studies in the lab An experiment was conducted providing only a limited amount of water available to each coated seed that demonstrated that the superabsorbent polymers held water in the coating, but freely provided the water to the germinating seed. In addition, the four tested SAP formulations were not phytotoxic to germination. The second lab study provided a simulated drought by allowing the growing media to dry after planting. In this scenario, free water was available for the SAP to rapidly swell, and hold the water around the coated seed resulting in germination in a drying environment. Previous published papers on SAP coated seed do not include studies under controlled environments.

Below, some specific comments:

ABSTRACT 

Point 1: Ln10. Be consistent and use either ‘superabsorbent’ or ‘super absorbent’. 

Response 1: Agree and edited throughout the manuscript as ‘superabsorbent’ 

Point 2: Ln14-15. ‘STR had greater water absorbency than other SAP materials’. Is this from methods or a result? I suggest deleting from the abstract unless it is a meaningful result and conclusions are developed. If it is a relevant result, you need to present it as such. 

Response 2: Determination of Water Holding Capacity of SAP Materials explained in section 2.1. of M&M and greater water holding capacity of STR is a meaningful part of the results.

Point 3: Ln18. Please, replace ‘and 4 SAP treatments’ with '...PAL, PAM, PAM+G and STR)'. Make sure to add the closing bracket.

Response 3: Abstract revised (line 9-24).

Point 4: Ln18. Delete ‘different’ and include the methods in brackets ‘(broadcast and broadcast with ranking).

Response 4: Abstract revised (line 9-24).

Point 5: Ln22. Please, replace ‘for the broadcast’ with ‘under broadcast planting’.

Response 5: Abstract revised (line 9-24).

Point 6: Ln22-23. Please, correct to ‘while STR had 31% greater plant stand than the noncoated control under broadcast with ranking planting’.

Response 6: Abstract revised (line 9-24).

Point 7: Ln23-24. Please, replace ‘selected hydrophilic seed coating’ with ‘selected SAP seed coating’.

Please, indicate if the differences in plant stands were statistically significant by adding significance level (e.g. P < 0.05).

Response 7: Abstract revised (line 9-24).

Point 8: Please, in the keywords, do not include words already used in the title such as seed coating, cover crop, superabsorbent polymers.

Response 8: Keywords edited Line 25: “seed technology; hydrophilic polymers; water deficit; germination

 INTRODUCTION

 Point 9: Ln29-33. This paragraph is identical to a presentation abstract publicly available and needs to be re-writing (https://ashs.confex.com/ashs/2021/meetingapp.cgi/Paper/34964).

Response 9: The paragraph paraphrased please see line 28-32.

Point 10: Ln37-38. Are references 5 and 6 related to semi-arid and arid environments? If not, please, replace with appropriate references.

Response 10: Reference 5 and 6, these studies took place in Middle-east Beirut, Lebanon and Egypt, Africa, which are considered semi-arid and arid regions, respectively.  

Point 11: Ln40. Reference 7 is not appropriate. Please, replace.

Response 11: Line 37-39 edited and Reference 7 replaced:

  1. Agaba, H.; Orikiriza, L.J.B.; Esegu, J.F.O.; Obua, J.; Kabasa, J.D.; Hüttermann, A. Effects of hydrogel amendment to different soils on plant available water and survival of trees under drought conditions. Clean Soil Air Water201038, 328–335.

METHODS

Point 12: You need to describe the differences between broadcasting and broadcasting with ranking. These methods yielded significantly difference seedling emergencies (Fig. 8b).

 Response 12: Line 182-184 Planting methods of Broadcast (B) and Broadcast and Raking (BR) described in M&M section 2.5.

RESULTS

Point 13: Fig. 9. Apparently, the significant comparisons were conducted all across the 12 treatments. However, this seems to be inappropriate because the planting method (broadcast vs. broadcast with ranking) might confound the results. If planting method is not included in the analysis as an interaction factor, I would suggest to perform ANOVA per trail and within planting method.

 Response 13: Authors agree with the suggestion of performing ANOVA for each field trial and method of planting separately. Also for clarification data of Figure 9 presented as Table 2.

Table 2. Percent plant stand count of 6 different treatments of two planting methods (B and BR) at three planting dates in the field (B = Broadcasting and BR= Broadcasting + Raking).

Stand count %

Trial 1

Trial 2

Trial 3

Treatments

B

BR

B

BR

B

BR

Control

6 c

13 bc

22 c

40 c

56 cd

70 bc

0% SAP

7 c

13 bc

26 bc

43 bc

51 d

70 bc

PAL

9 bc

15 B

30 ab

44 bc

65 ab

76 ab

PAM

6 c

13 bc

24 bc

45 ab

58 cd

73 abc

PAM+G

10 ab

13 bc

34 a

47 ab

62 bc

79 a

STR

8 bc

20 a

24 bc

52 a

67 ab

79 a

Within a column, mean values with different letters are statistically different (P < 0.01).

REFERENCES

Point 14: Please, be consistent and either include the doi link for all citation or don’t such as in reference 2: ‘https://doi.org/10.1007/BF00750062’. I suggest to include it in all references.

Response 14: Agree and doi links included for all applicable references.

---------------------------------------------------------------------------------------------------------------------

Response to reviewer #3: Authors of this manuscript appreciate the positive feedback from Reviewer #3

Reviewer #3

Comments and Suggestions for Authors

The methodology is easy to follow and the results are clearly presented.

Reviewer 3 Report

The methodology is easy to follow and the results are clearly presented.

Author Response

Response to reviewer #3: Authors of this manuscript appreciate the positive feedback from Reviewer #3

Reviewer #3

Comments and Suggestions for Authors

The methodology is easy to follow and the results are clearly presented.

Round 2

Reviewer 1 Report

The study is worthy for solving germination of seeds of cover crops especially in water stressed ecologies. The authors have made excellent story while preparation of the manuscript. The data in Table, figures and in graphics are appealing and relevant. However in germination and crop establishment more data may be included for better results. Like The water content of seeds and rate of hydration may be estimated , Timson’s index of germination velocity and also somem of the data n growth attributes may be included for better understanding on crop growth due to use of SAP. Overall, the manuscript is good, very well prepare and nicely explained the results and discussion. Also the study targets very important problem of seed germination in cover crops. 

Author Response

Reviewer #1 round 2

Comments and Suggestions for Authors

The study is worthy for solving germination of seeds of cover crops especially in water stressed ecologies. The authors have made excellent story while preparation of the manuscript. The data in Table, figures and in graphics are appealing and relevant. However in germination and crop establishment more data may be included for better results. Like The water content of seeds and rate of hydration may be estimated (For the rate of hydration estimation please see Video S1 of supplementary materials), Timson’s index of germination velocity (see below Line 176-181 & 403-407) and also some of the data on growth attributes may be included for better understanding on crop growth due to use of SAP. Details of the methods used are in lines 170 to 175. Overall, the manuscript is good, very well prepare and nicely explained the results and discussion. Also the study targets very important problem of seed germination in cover crops. 

Response to reviewer #2: We sincerely appreciate all valuable in-depth comments and suggestions, which helped us to improve the quality of the manuscript (In manuscript file track changes and yellow highlighted sections are our response to reviewer #1). 

Line 9 edited as “threaten”

Line 16 unit added “200 g water/g of polymer”

Line 88 edited as “2.1. SAP Materials and Determination of Water Absorbency of SAP Materials

Line 95 edited as “water absorbency”

Line 127: 2.3. Determination of Flowability of SAP Coated Seeds

Line 162-168 edited and written in one paragraph

Line 176-181: Germination velocity estimated by using Timson's index [26] was calculated for each treatment as follows.

Where, ?? is the cumulative germination percentage in time interval ? and ? is the total number of time intervals.

Line 182 edited as “2.5. Field Studies of Coated Seeds

Line 221 edited as “3.1. Determination of Water Absorbency of SAP Materials”

Line 306 “3.3. Seed Flowability of the SAP Coated Seeds”

Line 362 and 364 edited as “water absorbency”

Line 403-407: The Timson’s index [26] for nontreated Control, 0% SAP, PAL, PAM, PAM + G, and STR germination velocity was 169, 118, 282, 288, 283, and 284 respectively, where greater values have higher and faster germination. Collectively, all 2% SAP formulations had similar Timson’s index values ranging from 282 to 288 compared to the non-SAP checks.

Line 448: 3.5. Field Studies of Coated Seeds

Line 608 Reference 26

  1. Timson, J. New method of recording germination data. Nature 1965, 207 (4993): 216–217. doi: 10.1038/207216a0

Reviewer 2 Report

Dear Authors,

I'm very glad seeing the improvements in the manuscript and the meaningful results presented. The abstract conveys well the findings of this research and the manuscript became an original helpful piece for the scientific and the farming community with practical information and clear results.

very minimal changes might be addressed if you find appropriate:

-Table 2. Please, uncapitalize the ‘B’ used for Trail 1, BR, PAL

-Ln91. Please, start a new paragraph in ‘Volumetric liquid absorption…

-Figure 5d. I suggest to use uncapitalized letters to show statistical differences.

-Ln300. ‘Pawlicki et al (2019)…’ it seems you missed here to add the reference number which should be in [], please double check. Is it reference [31]?

With best regards,

Cristian Moreno Garcia

Author Response

Reviewer #2 round 2
Comments and Suggestions for Authors
Dear Authors,
I'm very glad seeing the improvements in the manuscript and the meaningful results presented. The abstract conveys well the findings of this research and the manuscript became an original helpful piece for the scientific and the farming community with practical information and clear results.
very minimal changes might be addressed if you find appropriate:
Response to reviewer #2: We wish to express our appreciation for your suggestions, corrections, and positive feedback which have greatly improved the manuscript. (In the manuscript file track changes and green highlighted sections are our response to reviewer #2)
 -Table 2. Please, uncapitalize the ‘B’ used for Trail 1, BR, PAL 
Authors prefer to use uppercase for abbreviations throughout the manuscript text, figures and tables.  
-Ln91. Please, start a new paragraph in ‘Volumetric liquid absorption…’ 
Edited, please see line 94
-Figure 5d. I suggest to use uncapitalized letters to show statistical differences. 
Edited, please see line 352-355 Figure 5d
-Ln300. ‘Pawlicki et al (2019)…’ it seems you missed here to add the reference number which should be in [ ], please double check. Is it reference [32]? 
Reference number added please see line 326 &330
